# Development of a detailed canine gait analysis method for evaluating harnesses: A pilot study

**Zsófia Pálya, Kristóf Rácz, Gergely Nagymáté, Rita M. Kiss** [ID]*

Department of Mechatronics, Optics and Mechanical Engineering Informatics, Budapest University of Technology and Economics, Budapest, Hungary

* rita.kiss@mogi.bme.hu

## Abstract

Dog harnesses are becoming more popular, with their large variety stemming from the idea that different dogs and scenarios require different types of harnesses. While their benefits over collars are self-explanatory, there is a lack of research on their effect on gait, and even the existing studies examine only a limited set of parameters. The goal of present study was to establish a method capable of quantifying canine gait in detail. Based on 3D motion capture, the developed method allows for the examination of 18 joint angles and 35 spatio-temporal parameters throughout multiple gait cycles, and can be used to analyze canine movement in detail in any kind of scenario (e.g. comparing healthy and lame dogs, or measuring the effect of training). The method is presented through the measurement of how different harnesses affect walking kinematics compared to free (unleashed) movements. Four dogs with varying body sizes and breeds and multiple types of harnesses were included. Marker data was filtered using a zero-lag 6th order Butterworth-filter with a cutoff frequency of 20 Hz. The normality of the spatio-temporal and joint range of motion parameters was tested using the Anderson-Darling test (p = 0.05), with most parameters passing in 60+% of test cases. Swing time and range of motion of the sagittal aspect of spinal angle at T1 vertebrae failed more regularly, both resulting from the measurement setup rather than the actual parameters being not normally distributed. Two-sample Kolmogorov-Smirnov tests (p = 0.05) were used to compare each parameter's distribution between cases, showing that most parameters are significantly altered by the harnesses in about 2/3rd of the cases. Based on the results, there's no absolute superior harness, however, it is possible to select the best fit for a specific dog and application, justifying their large variety.

## Introduction

Gait analysis is a well-established and objective technique to assess normal and abnormal gait accurately, identifying characteristic features of specific gait abnormalities. Besides the wide range of application in human motion analysis (e.g. medicine, sport, rehabilitation), gait evaluation can also be used in veterinary medicine, e.g. the diagnostics of lameness in horses or

**Data Availability Statement:** All relevant data are within the paper and its Supporting information files.

**Funding:** The research reported in this paper and carried out at BME has been supported by the

Hungarian Scientific Research Fund (OTKA), grant number: K135042 (http://nyilvanos.otka-palyazat. hu/index.php?menuid=930&num=135042&lang= EN), and the National Research, Development and Innovation Fund (TKP2020 NC, No. BME-NCS) based on the charter of bolster issued by the National Research, Development and Innovation Office under the auspices of the Ministry for Innovation and Technology, Hungary. The three harnesses from Julius-K9® were provided free of charge by the manufacturer. The funders had no role in study design, data collection and analysis, decision to publish, or preparation of the manuscript.

**Competing interests:** The authors have declared that no competing interests exist.

dogs [1–3]. The measurement techniques within gait analysis can be divided into two primary groups: kinetic and kinematic. Kinetic measurements mostly focus on the forces acting between the foot and the ground (ground reaction forces), measured with force platforms or a baropodometric system, while kinematic measurements record the position and orientation of body segments or bony landmarks, from which the linear and angular velocities, accelerations and the joint angles are determined [4]. With precise measurement techniques, it is possible to quantify gait, so that different groups or populations can be compared numerically.

Veterinary science started to apply the techniques of human motion analysis to study the lameness of horses [1]. In canine science, quantitative gait analysis appeared in the late 20th century [5–7]: the first application was the examination of the resultant truncal and limb transmission force; secondly, joint-specific musculoskeletal functions were analysed with 2-dimensional gait analysis [8]. Nowadays, three-dimensional motion capture is the most common method for examining both human and animal movement patterns. Besides lameness, these various methods may be suitable for analysing different pathologies, such as gait patterns before and after therapeutic treatments or a distinct environmental effect on gait [8–12].

Instead of the regular neck collar, harnesses are becoming more popular, partly due to the emerging canine sports activities, and partly because it can provide better control over dogs, even with the presence of behavioural issues. There are many styles of body harnesses, which can generally be classified as restrictive or non-restrictive. Non-restrictive harnesses feature a Y-shaped chest strap across the body above the scapula, while restrictive harnesses have a strap coming across the chest, crossing the body at, or below the scapula [13]. The large variety of harnesses is the result of an empirical idea that different harnesses are better for certain dogs or scenarios than others. While the benefits of a harness over the neck collar seem self-explanatory, the downsides are more subtle. Most owners and trainers do not observe the gait alternations caused by the harness. These may contribute to long-term repetitive strain, leading to or predisposing an injury [9]. Based on Carr et al., during irregular activates (i.e. training or competing), an objective gait analysis should be performed with the dog wearing the harness to identify any gait alterations [14]. However, according to Blake S. et al., only three full papers were written about examining the biomechanical effects of harness and neck collar use in dogs until September 2019 [15]. Lafuente et al. studied the differences between restrictive and non-restrictive harnesses on shoulder extension, and both harnesses showed significantly decreased joint angle range of motion (ROM) [13]. These conclusions agree with those by Carr et al. in a conference abstract paper, based on examining five different harnesses using a pressure sensitive walkway [14]. Peham et al. investigated the movement of the spine [16] and the pressure distribution under three different types of harnesses designed for guided dogs [17]. They found that spinal motion changed significantly, and calculated forces were greater under the trunk strap. Furthermore, they concluded that there were measurable differences between three types of harnesses. One more study published in March of 2020 investigated the truncal motion of dogs in service vests [18], where they found that the vest has significantly changed the truncal motion of the animal.

A fundamental limitation of previous studies is the low complexity of analysis. They rarely examine more than a few kinematic parameters (e.g. stance time percentage, stride length, and step length [14]) or only calculate minimum, maximum and ROM for joint angles at most (e.g. maximum shoulder extension [13]), instead of describing the joint angle over the full gait cycle. This pilot study's first goal is to establish a measurement method capable of quantifying canine gait in it's great complexity, to support long term research, be it the effects of different harnesses have on movement, or any other research related to dog motion. This measurement method should be able to determine spatio-temporal parameters of all four limbs (35 scalar parameters) and the joint angles of the major joints and spinal angles (18 joint angles

throughout the whole gait cycle, as well as scalar ROM parameters) captured through multiple gait cycles. We hypothesised that the method will be suitable for all sizes of dogs and different harness types.

## Methods

### Study design

Four dogs could be recruited for this pilot study whose owners agreed on training their dogs for treadmill walking. This included small, medium and large body sizes with varying breeds, to test the suitability of the method for different circumstances. During the pre-measurements period, the owners trained the dogs to walk on a treadmill for 4–6 weeks (depending on progress), one or two 15 minute sessions per week.

On measurement day, the dog and its owner were provided a 20 minutes accommodation session to become familiar with the treadmill and the environment [2]. Each owner determined the belt velocity during the habituation process so that it seemed comfortable for their dogs. This selected speed was then kept constant for that dog throughout the recordings. Gait patterns were identified by eye by an expert (detailed description of each gait pattern can be found in [19]). The measurements were performed on a ProFitness L150 treadmill (Argos, Milton Keynes, UK) which has a 123 cm long and 40 cm wide belt and has a maximum belt velocity of 12 km/h. During the measurements, the dog's owner was squatting in front of the treadmill and periodically praised the dog with treats to maintain a continuous walk (Fig 1). The processed section of the trial was a selected homogeneous gait section between treats, usually containing 20 to 30 gait cycles. The movement patterns of each dog were recorded by applying three different types of measurement conditions. Firstly, the reference motion was captured without the dogs wearing any harness. Next, the motion was recorded with the different harnesses without a leash attached, and lastly while wearing each harness with a leash attached. The number of recorded sessions was determined by the number of available harnesses, summarised in Table 1. The order of measurements was the same for all dogs.

**Dog studied.** Four clinically healthy short-haired dogs aged between 3 and 5 years participated in the study, whose anthropometric data is summarised in Table 1. All dogs participated in the measurement with their owners, who have given their written consent to participate in the experiment after they were informed verbally about all aspects (e.g. hazardous situations, instrumentation, use of media contents, etc.). The study was approved by the National Science and Research Ethics Committee (Hungary) (21/2015).

**Harnesses studied.** An important consideration was to select both restrictive and non-restrictive (Y-type) harnesses into our study, which are commercially available to most owners in the country and manufactured in several sizes. In this research, three harness types were

**Table 1. Participating dogs and used harnesses.** Gait patterns were identified by eye by an expert. Detailed description of each gait pattern can be found in [19].

| ID | Breed | Weight (kg) | Harnesses studied | Gait pattern | Treadmill velocity (m/s) |
|---|---|---|---|---|---|
| Dog 1 | Bullterrier | 26 | Julius-K9®power, Julius-K9®IDC, Julius-K9®Duo-Flex | pace | 0.91 |
| Dog 2 | Bullterrier | 16 | Julius-K9®power, Julius-K9®IDC, Julius-K9®Duo-Flex | amble | 0.92 |
| Dog 3 | Yorkshire terrier | 3 | Julius-K9®power, Julius-K9®IDC | walk | 0.8 |
| Dog 4 | Beagle-labrador mixed breed | 26 | Julius-K9®power, Julius-K9®IDC, Julius-K9®Duo-Flex, Fressnapf own-branded | amble | 1.05 |

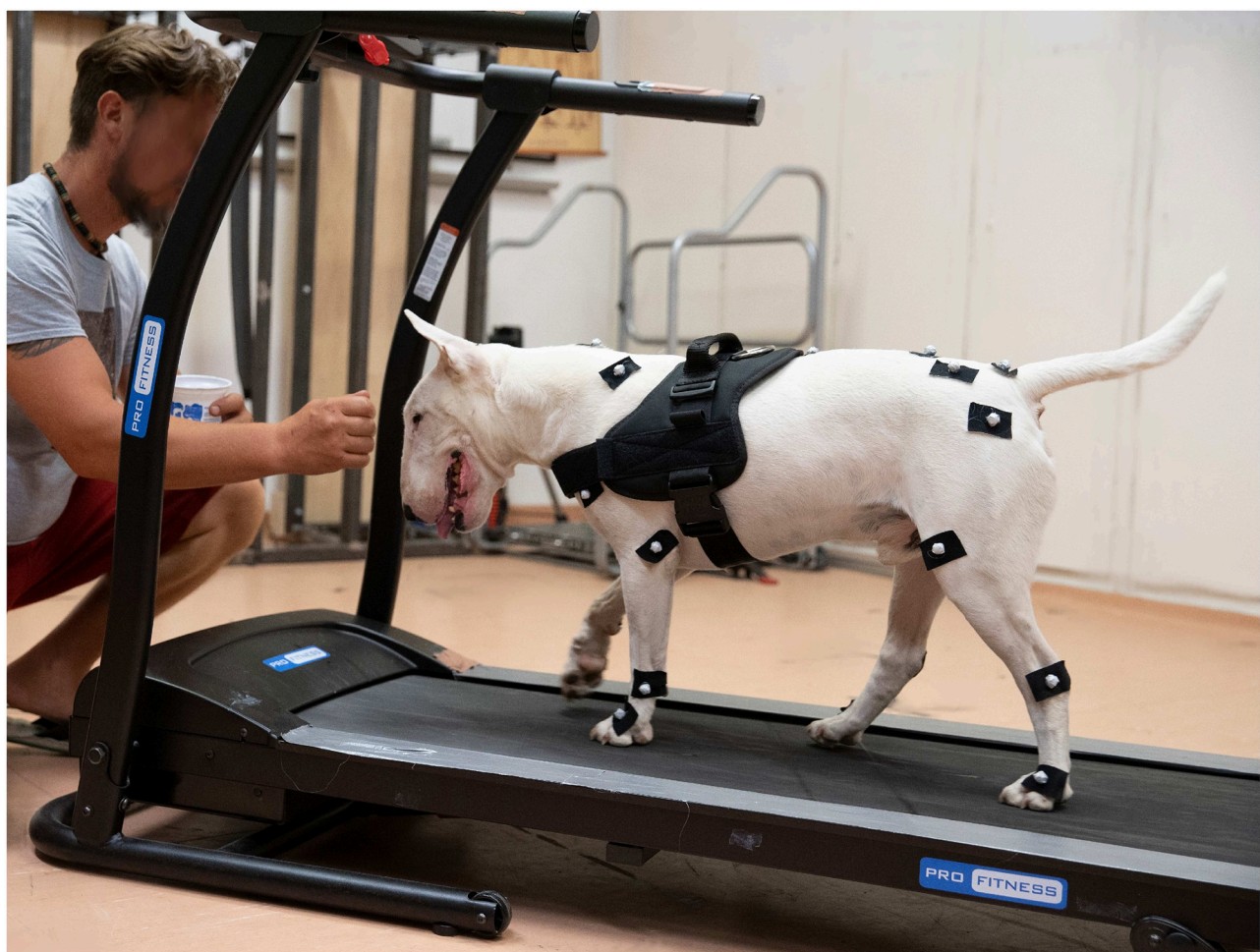

**Fig 1. Measurement arrangement.**

studied which were manufactured by Julius-K9®and one which was provided by the dog's owner (only in case of Dog 4). Two of them were restrictive harnesses, namely the Julius-K9®power harness (Fig 2c) and the Julius-K9®IDC harness (Fig 2b). The Julius-K9®Duo-Flex harness (Fig 2a) was a non-restrictive one. These harnesses were manufactured specifically for this research without the light-reflexive materials, which could interfere with motion capture measurements (this did not affect the mechanical properties of the harnesses). In the case of Dog 4, a generic non-restrictive harness was also included (Fig 2d). This harness was a pet store (Fressnapf GmbH, Krefeld, Germany) bought own-branded product. Since it was made with light-reflective patterns, the reflective surfaces was covered with adhesive tapes during the measurements. No regular neck collar was included in the study, as it was determined to not interfere with marker placement as harnesses could, and therefore the developed measurement should be also applicable to collars without issues. All the dogs were previously accustomed to wearing the harnesses. Table 1 also shows which dog was measured with which harnesses. A total of 28 trials was recorded, including all free, harness only, and leashed trials.

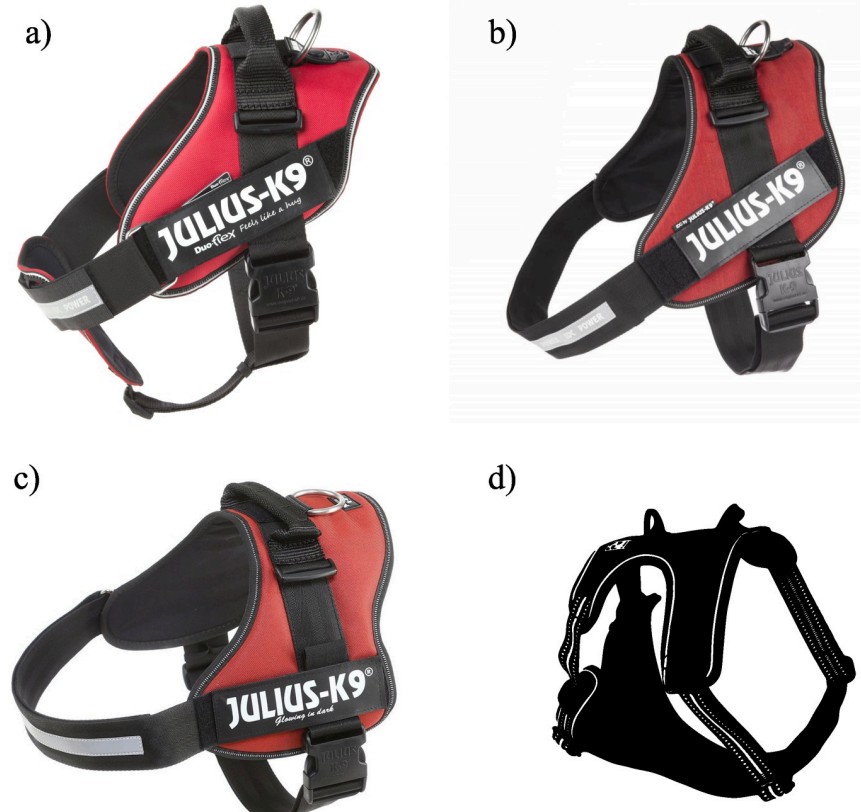

**Fig 2. Studied Julius-K9® harnesses.** a) Julius-K9®Duo-Flex harness; b) Julius-K9®IDC harness; c) Julius-K9®power harness; d) sketch of the Fressnapf own-branded harness. Note, that the Julius-K9®harnesses were manufactured without the light-reflexive materials. Pictures a), b) and c) are reprinted from https://julius-k9.com/en/ under a CC BY license, with permission from Julius-K9®, original copyright (1997–2020).

## Measurement setup

The measurement system was an OptiTrack (NaturalPoint, Corvallis, Oregon, USA) three-dimensional optical motion capture system consisting of 18 Flex13 cameras and the Motive v1.10.3 software [20]. The software is responsible for the coordinated operation of the 18 cameras and for recording the 3-dimensional position of markers. This measuring system's accuracy is sub-millimetre (3D MeanErr: 0.537±0.016 mm), and the measurements were carried out at a sampling frequency of 120 Hz [21]. The cameras cover a 4x2.5 m measuring area, the treadmill being placed in the middle of it. The measurements were carried out at the Department of Mechatronics, Optics and Engineering Informatics at Budapest University of Technology and Economics, Hungary.

Based on Hogy et al., twenty-five infra reflective markers were placed on the dog's specific anatomical landmarks (Fig 3) [22]. The same marker-set arrangement was used for each measurement. On the thoracic limbs, markers were placed over the distal lateral aspect of the fifth metacarpal bone (FR5 and FL5), the ulnar styloid process (FR4 and FL4), the lateral epicondyle of the humerus (FR3 and FL3), the greater tubercle of the humerus (FR2 and FL2), and the dorsal aspect of the scapular spine (FR1 and FL1). On the pelvic limbs, markers were placed over the distal lateral aspect of the fifth metatarsal bone (BR5 and BL5), the lateral malleolus of the fibula (BR4 and BL4), the lateral femoral condyle (BR3 and BL3), the greater trochanter of

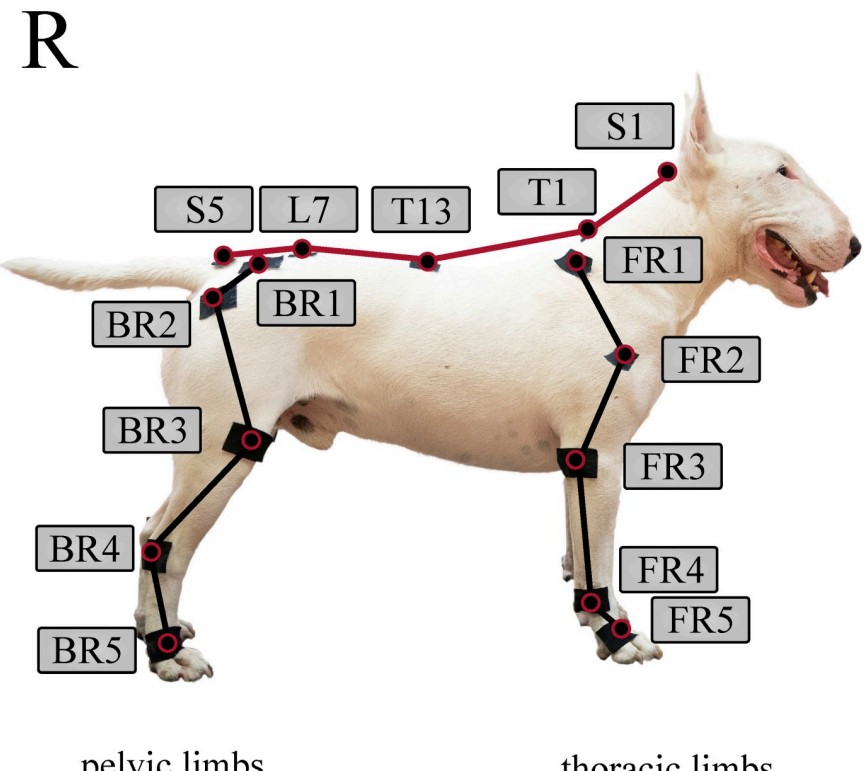

**Fig 3. Marker-set of 25 reflexive markers.** Note, that markers on the left side are not shown. Illustrative purpose only, the marker placement procedure was carried out according to Hogy et al. [22].

the femur (BR2 and BL2), and the iliac crest (BR1 and BL1). On the spine, markers were placed over the sacral apex (S5), the dorsal spinous process of vertebra L7, the dorsal spinous process of vertebra T13, the dorsal spinous process of vertebra T1, and the occipital protuberance (S1). The markers were fixed on the dogs using adhesive tape. Each participating dog was short-haired, thus their fur did not influence marker placement or induced unnecessary marker movement. In cases where a harness covered the location of an anatomical landmark (FR2-FL2), the marker was placed on top of the harness, as close to the anatomical landmark as possible.

### Post-processing

**Exporting data.**    A technician first processed the recorded marker data in Motive as follows: markers were manually labelled according to the used marker-set (Fig 3) for each recording. Next, a section of homogeneous gait between receiving treats was selected for each trial, and exported into a text file containing metadata of the measurement in a header—like frame rate and total number of frames—and the marker position data for each frame. For all further calculations, MATLAB (R2020b) was used [23].

**Filtering.**    Filtering marker data with a low-pass Butterworth filter ($f_c$ = 5–6 Hz) is a long-accepted norm in human gait analysis [24–26]. However, when inspecting positional marker data before and after applying this filter, it is apparent that it is not suitable for canine measurements, especially for smaller dogs. Larger movements (typically in the cranial-caudal direction) have good signal-to-noise ratio to begin with, and thus are not significantly affected by the

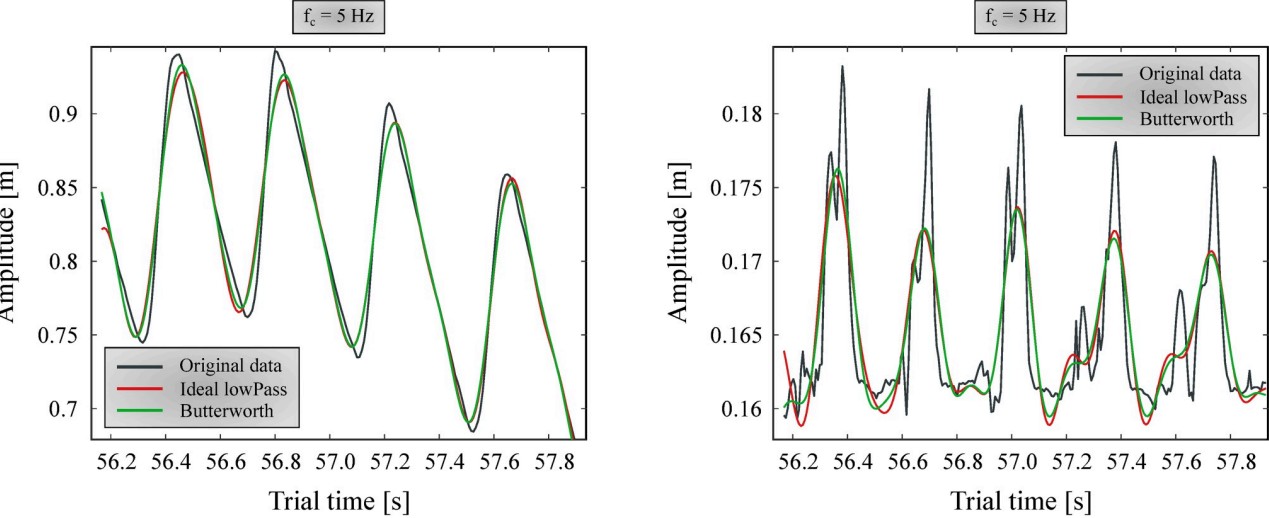

**Fig 4. Position data of FR5 marker of the smallest dog filtered with $f_c$ = 5 Hz.** Ideal low-pass and 6th order Butterworth filters are shown in figure. Left: forward direction; Right: vertical direction.

filtering (Fig 4 left). However, the smaller components (vertical direction) have useful data beyond 5 Hz, and their signal is significantly distorted, especially on smaller dogs (Fig 4 right).

A 6th order zero-lag (using the *filtfilt* method for eliminating phase delay [23]) Butterworth filter and an 'ideal' low-pass filter were compared with multiple cutoff frequencies ($f_c$). The 'ideal' low-pass filter was achieved by applying fast Fourier transform (FFT) to the data, removing the components on frequencies above $f_c$, and converting the resultant spectrum back to a time series with inverse FFT. Based on observing multiple cases, it was found that resulting information loss affected the smallest dog's marker positions the most. Accordingly, $f_c$ was tuned to provide good results for the smallest dog, assuming that noise is independent of the dog measured, and occupies the same frequencies for all trials. After examining multiple $f_c$ values, 20 Hz was determined to be appropriate, where no relevant information seemed to be lost due to filtering. A forward and vertical landmark position component before and after filtering with this cutoff frequency can be seen on Fig 5.

Both for $f_c$ selection and the actual processing, non-harmonic components are removed before applying the filters by fitting a trend line to the marker position component and subtracting it. Afterwards, the trend is re-added to the position data. We found that the Butterworth filter could more smoothly follow the original signal, whereas the 'ideal' filter had trouble matching the signal's endpoints, introducing artefacts at the beginning and end. Thus, the selected filter was a 6th order Butterworth filter at $f_c$ = 20 Hz (Fig 5, green lines), which was applied for every maker position component.

**Gait cycle segmentation.** In the next step, the recording was segmented into complete gait cycles from which the spatio-temporal parameters can be calculated. Heel strike and toe-off events for all limbs have to be determined. Similarly to human gait, heel strike occurs when the foot is farthest forward compared to the hip/shoulder [27], while toe-off occurs while the foot is the furthest behind the given joint. The corresponding frame numbers are found by calculating the feet-shoulder/hip joint distances and finding the peaks with the *findpeaks* function [23]. The back right limb's heel strike (based on marker BR5, see Fig 3) was chosen as the separating points of the gait cycles.

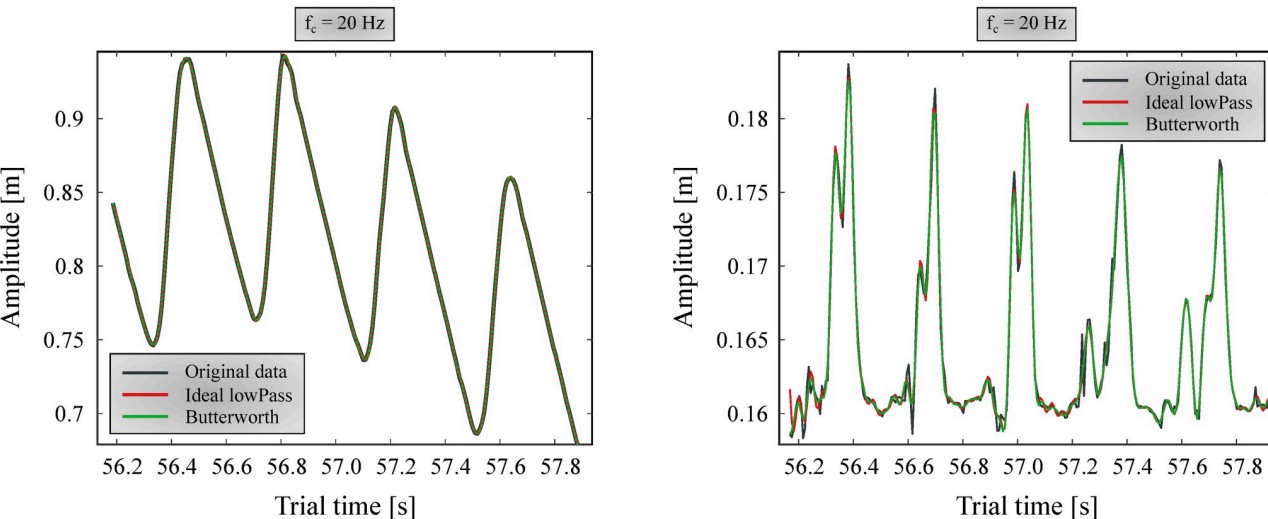

**Fig 5. Position data of FR5 marker of the smallest dog filtered with $f_c$ = 20 Hz.** Ideal low-pass and 6th order zero-lag Butterworth filters are shown in figure. Left: forward direction; Right: vertical direction.

## Measured and calculated parameters

**Paw parameters.** A common parameter to represent the change in the cyclic forward movement is the path traced by selected points, such as the head or the centre of mass, but any measured point can be drawn. Paw movement (plots of each limb's fifth marker in the sagittal plane) was chosen as a way to represent the cyclic forward movement of the animal. Other common points used to represent the forward movement is the centre of mass or the head, but centre of mass is difficult to calculate with the given marker set, and head movements were found to be very inconsistent due to the measurement setup, as the head position was not 'fixed' compared to the body (the dogs were moving their head around as to better observe things).

**Joint angle calculation.** The 2D projections of joint angles are calculated for each data frame. Spine angles in the horizontal plane at T1, T13 and L7 vertebrae are defined on Fig 6a. Joint angles in the sagittal plane include the spinal angles at the T1, T13 and L7 vertebrae, and the upper (shoulder/hip), middle (elbow/stifle) and lower (left and right carpal/hock) joint angles for all four limbs (Fig 6b). Joint angles are calculated by taking the two coordinates of the three defining anatomical landmarks in the given plane for each data frame and calculating the angle between the connecting vectors. Detailed descriptions of the joint angles are given in Table 2.

The calculated joint angle time series are segmented according to the back right foot's heel strike (the ends of gait cycles) resulting in one time series for a given joint angle per gait cycle. These are then re-interpolated to 101 equally spaced points giving the joint angle value at each percent of the gait cycle, from 0 to 100%. Multiple time series for a given joint can be represented with an average joint angle and a 95% confidence band.

**Spatio-temporal parameters.** With knowing the heel strike and toe-off locations, as well as the set speed of the treadmill (noted from the treadmill's display), the spatio-temporal parameters of each gait cycle can be calculated (Table 3). ROM for every gait cycle for each joint angle is also calculated and evaluated the same way as spatio-temporal parameters. In total, 18 joint angle curves (detailed in Fig 6 and Table 2), 4 plots of the paw movement in the

a) horizontal plane:

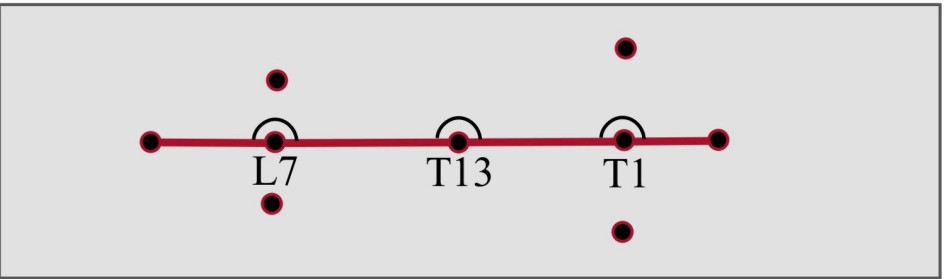

b) sagittal plane:

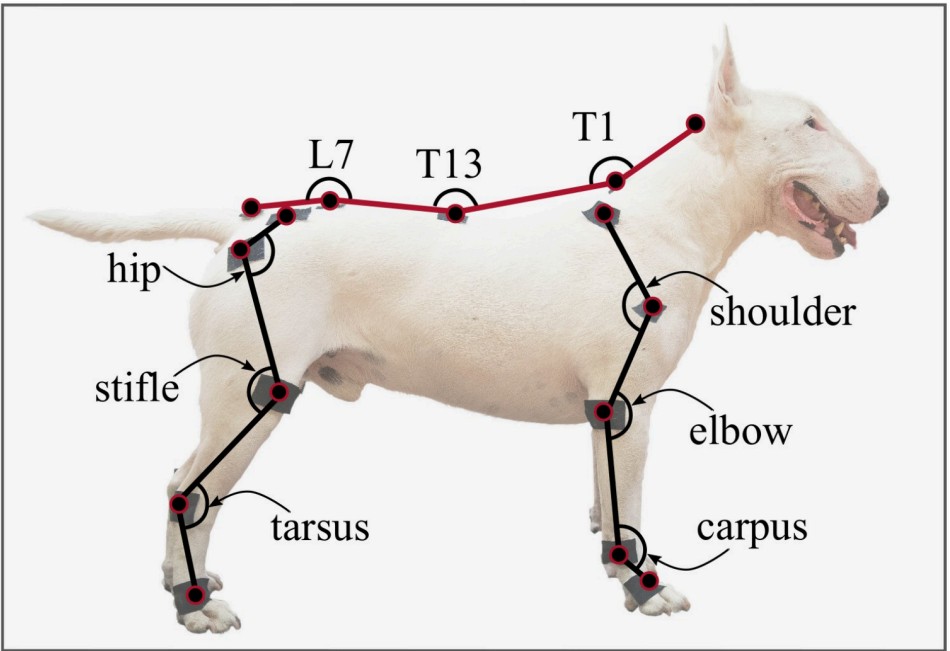

**Fig 6. Joint angles in the horizontal and sagittal plane.**

sagittal plane and 53 scalar parameters (35 spatio-temporal parameters detailed in Fig 7 and Table 3 and 18 joint angle ROM parameters) can be given for each gait cycle.

### Evaluation

**Per-case report.** The goal of the presented processing method is to allow for a thorough analysis and comparison of gaits. To help this, automatically generated reports were created that allows for the visual examinations of all parameters. The joint angles and the shape of the path traced by the paws during movement are represented as plots, while all of the spatio-temporal and ROM parameters are given in violin-like charts. Multiple trials can be displayed on the same plot, providing a basis for evaluating how a specific circumstance (free walk, free

**Table 2. Description of the calculated joint angles.**

| Joint angle name | Description | Plane |
|---|---|---|
| **T1 spinal angle** *(horizontal aspect)* | angle of T1-S1 and T1-T13 | dorsal/ horizontal |
| **T13 spinal angle** *(horizontal aspect)* | angle of T13-T1 and T13-L7 | dorsal/ horizontal |
| **L7 spinal angle** *(horizontal aspect)* | angle of L7-T13 and L7-S5 | dorsal/ horizontal |
| **T1 spinal angle** *(sagittal aspect)* | angle of T1-S1 and T1-T13 | sagittal |
| **T13 spinal angle** *(sagittal aspect)* | angle of T13-T1 and T13-L7 | sagittal |
| **L7 spinal angle** *(sagittal aspect)* | angle of L7-T13 and L7-S | sagittal |
| **Shoulder angle** *(upper front joint angle)* | angles of FR2-FR1 and FR2-FR3 (right side); angles of FL2-FL1 and FL2-FL3 (left side) | sagittal |
| **Hip angle** *(upper back joint angle)* | angles of BR2-BR1 and BR2-BR3 (right side); angles of BL2-BL1 and BL2-BL3 (left side) | sagittal |
| **Elbow angle** *(middle front joint angle)* | angles of FR3-FR4 and FR3-FR2 (right side); angles of FL3-FL4 and FL3-FL2 (left side) | sagittal |
| **Stifle angle** *(middle back joint angle)* | angles of BR3-BR4 and BR3-BR2 (right side); angles of BL3-BL4 and BL3-BL2 (left side) | sagittal |
| **Carpus angle** *(lower front joint angle)* | angles of FR4-FR5 and FR4-FR3 (right side); angles of FL4-FL5 and FL4-FL3 (left side) | sagittal |
| **Tarsus angle** *(lower back joint angle)* | angles of BR4-BR5 and BR4-BR3 (right side); angles of BL4-BL5 and BL4-BL3 (left side) | sagittal |

**Table 3. Spatio-temporal parameters.**

| Joint angle name | Description | Dim |
|---|---|---|
| **Stride time / Cycle time** | length of the gait cycle in time | s |
| **Cadence** | number of steps per minute calculated as 4 · (60/Stridetime) | steps / time |
| **Speed** | speed of the forward movement of the animal, calculated as the average of the back and front stride distance divided by stride time | m/s |
| **Stride distance** *(back and front)* | distance from one heel strike to the next | m |
| **Walking base** *(back and front)* | distance of the given limb's heel strike form the opposite side limbs heel strike in the direction of movement | mm |
| **Step distance** *(for all four limbs)* | distance of the given limb's heel strike form the opposite side limbs heel strike in the direction of movement | m |
| **Step height** *(for all four limbs)* | the range of the vertical component of the 5th marker on the given limb (how high the dog lifts it's paw) | mm |
| **Swing time** *(for all four limbs)* | the time when a given limb is not in contact with the ground (time from toe-off to heel strike) | s |
| **Swing ratio** *(for all four limbs)* | the ratio of the swing time compared to the full cycle time | - (%) |
| **Stance time** *(for all four limbs)* | the time when a given limb is in contact with the ground time from heel strike to toe-off) | s |
| **Stance ratio** *(for all four limbs)* | the ratio of the stance time compared to the full cycle time | - (%) |
| **Paw travel distance** *(for all four limbs)* | distance that the foot travels during one stride on the treadmill | m |

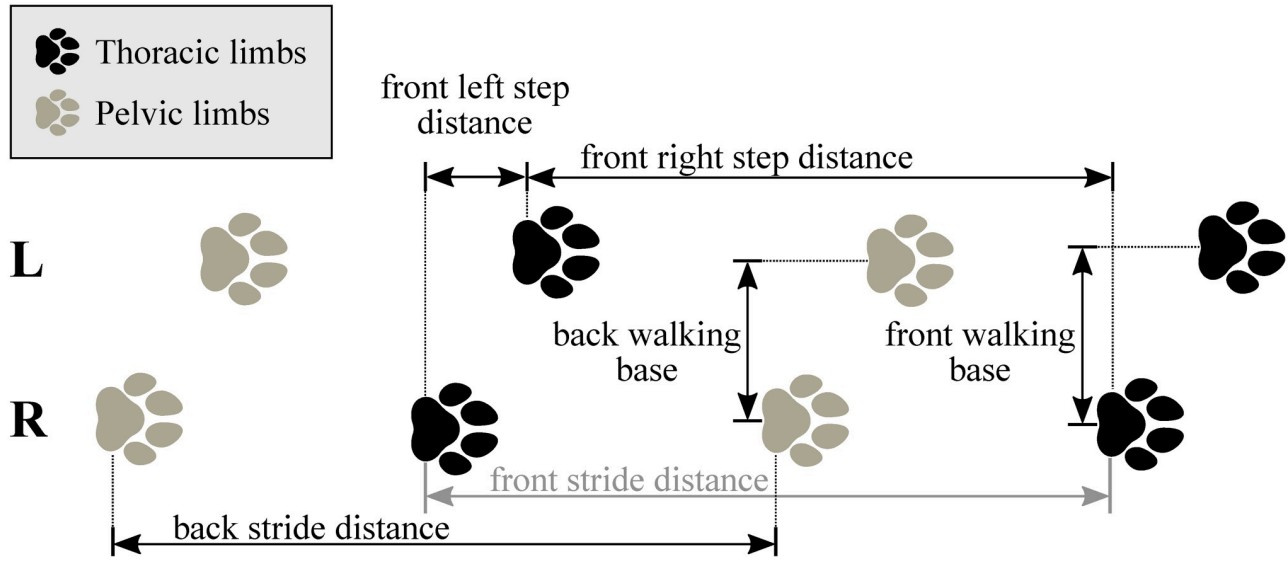

**Fig 7. Distance type parameters in the horizontal plane.**

walk wearing the harness and leashed walk wearing the harness) affects a particular dog's motion. An example of a report like this is given can be seen in Fig 8.

These reports provide useful insights into individual cases, e.g., selecting the best harness for a working dog. The used method could also be applied to other canine gait analysis cases e.g. comparing movement before and after treatment for lameness.

**Statistical analysis.** Individual reports do not provide insight into the overall effects of harnesses, which is the test case of this pilot study. For this purpose, statistical evaluations were used. First, we tested for normality of the spatio-temporal and ROM parameter distributions across gait cycles in the different measurement cases (the free, harness wearing and leashed scenarios) with the Anderson-Darling test (H0: The distribution is from a normal distribution, $\alpha = 0.05$). Parameters that do not pass the normality test more often can indicate problem areas for the specific dog or case. Next, each parameter's distributions for each dog-harness combination for the cases of reference vs harness, harness vs harness+leash, and reference vs harness+leash were tested for identical distributions. One test involves comparing the samples of one parameter collected over multiple gait cycles of the same dog in the two different trials. For this, the two-sample Kolmogorov-Smirnov test for identical distributions (H0: The two sample sets are from the same distribution, $\alpha = 0.05$) were used. This test can show if either the expected value or the parameter's variation has changed (it does not tell which one, however) and does not require the datasets to be normally distributed. In total 3 times 53 tests were needed per dog-harness combination.

Additionally, for all harness and harness+leash cases, root mean square (RMS) errors between the average joint angles were calculated as:

$$\text{RMS}_{trial} = \sqrt{\frac{\sum_{i=0}^{100} \left( \varphi_{ref,i} - \varphi_{trial,i} \right)^2}{101}}, \tag{1}$$

where trial represents a specific case (e.g. Dog 2 with the Julius-K9®power harness and leash), $\varphi_{ref,i}$ is the average joint angle of the reference trials of the given dog at the i-th integer percent of the gait cycle, and $\varphi_{trial,i}$ is the average joint angle of the selected trial of the given dog at the

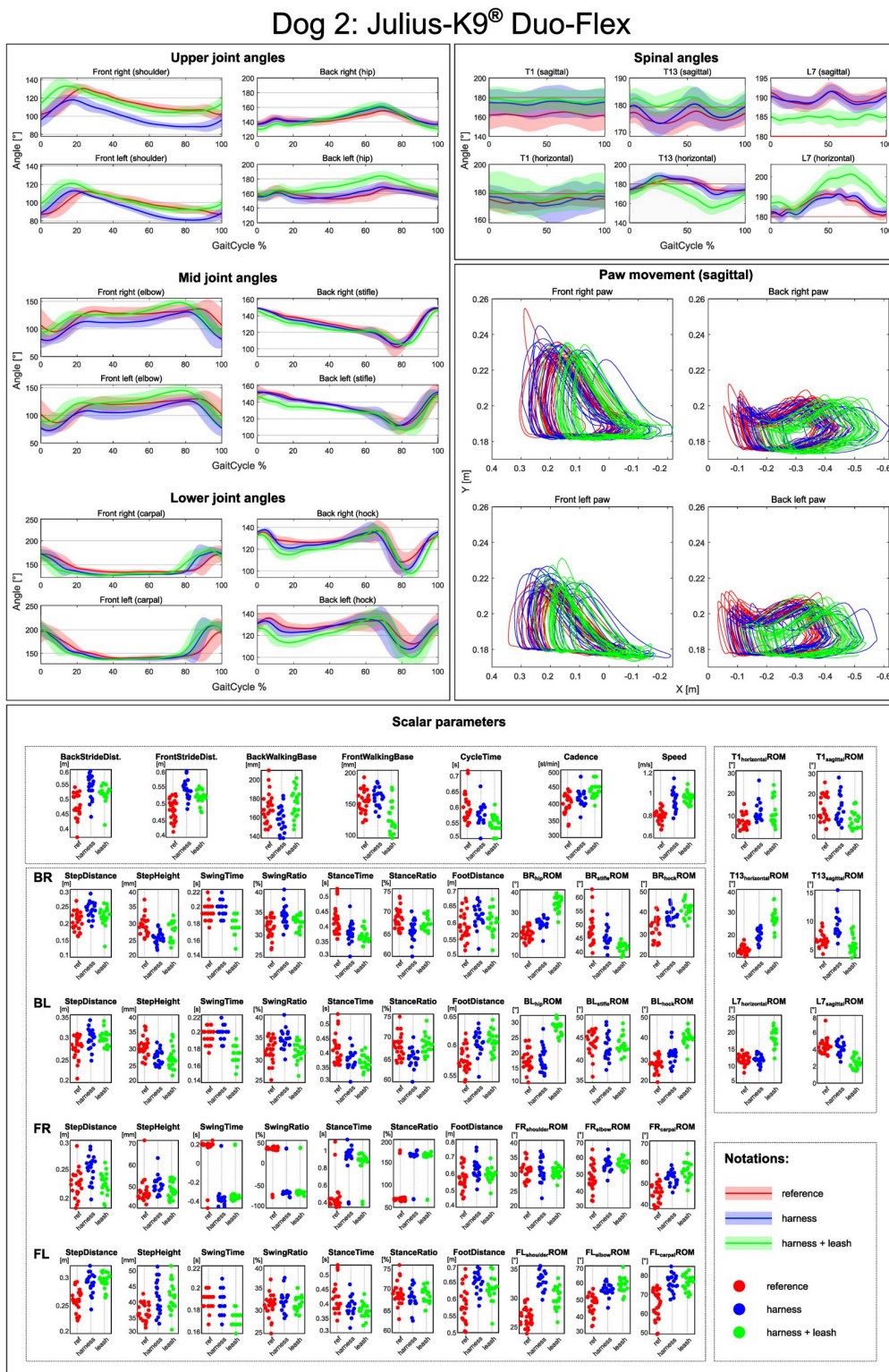

**Fig 8. An example of a report comparing free movement with harness and harness + leash scenarios.**

i-th integer percent of the gait cycle. This value assigns a numerical score for how close a joint angle is to the reference.

## Results

Tabular results of the 53 calculated parameters for all cases can be found in S1 Table. The table of results for the Anderson-Darling normality test can be found in S2 Table. Fig 9 shows for each parameter the percentage of cases where the parameter passed the Anderson Darling normality test ($p > 0.05$). Meanwhile, Fig 10 shows the inverse relation: how many parameters from a given trial passed the normality test.

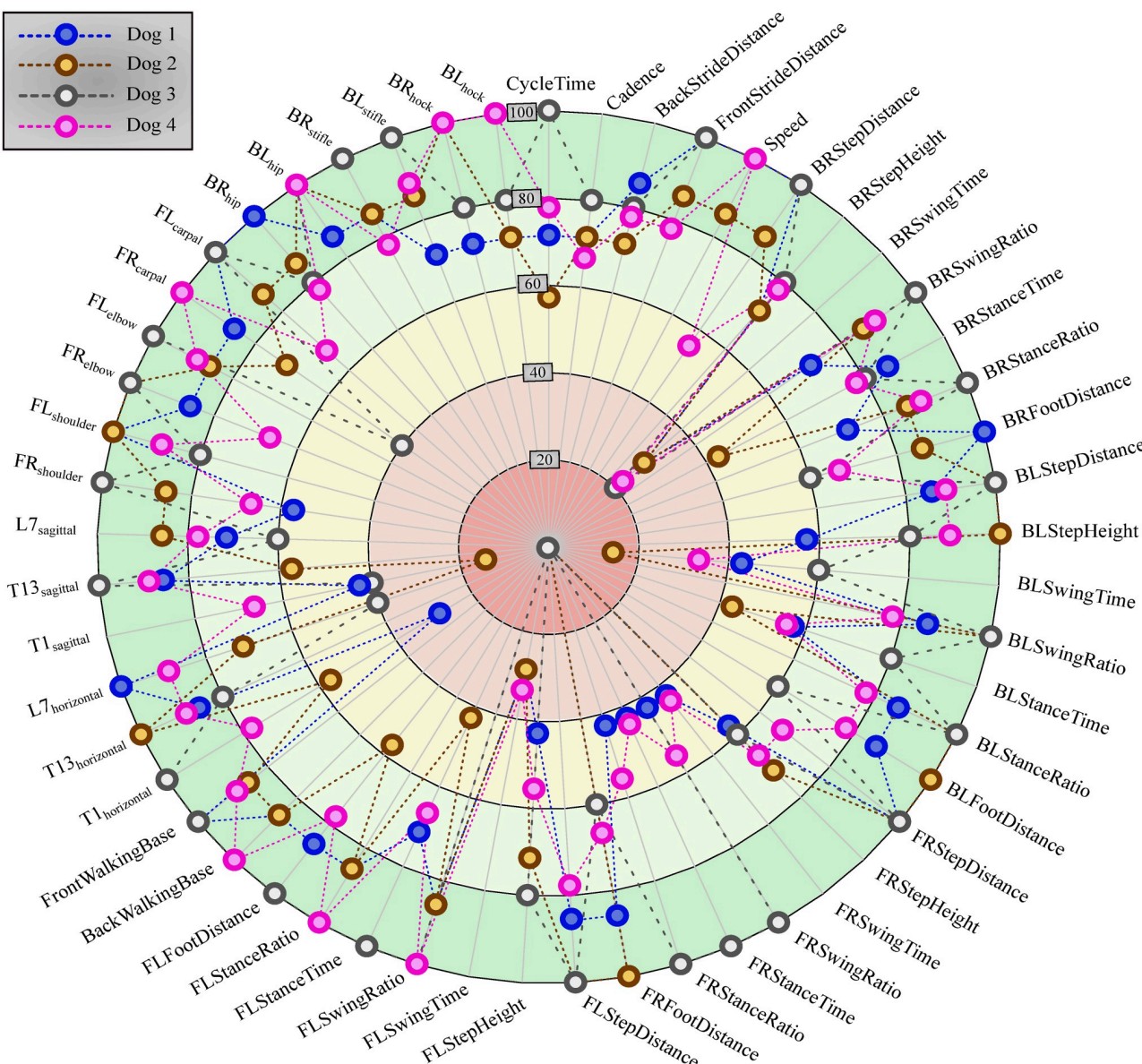

**Fig 9. Result of Anderson-Darling normality test for each parameter.** Percentage of cases (indicated by the distance from the centre) shows that a given parameter (spatio-temporal or joint range of motion, as named on the edge of the circle) passed the Anderson-Darling normality test ($p > 0.05$) (28 cases in total).

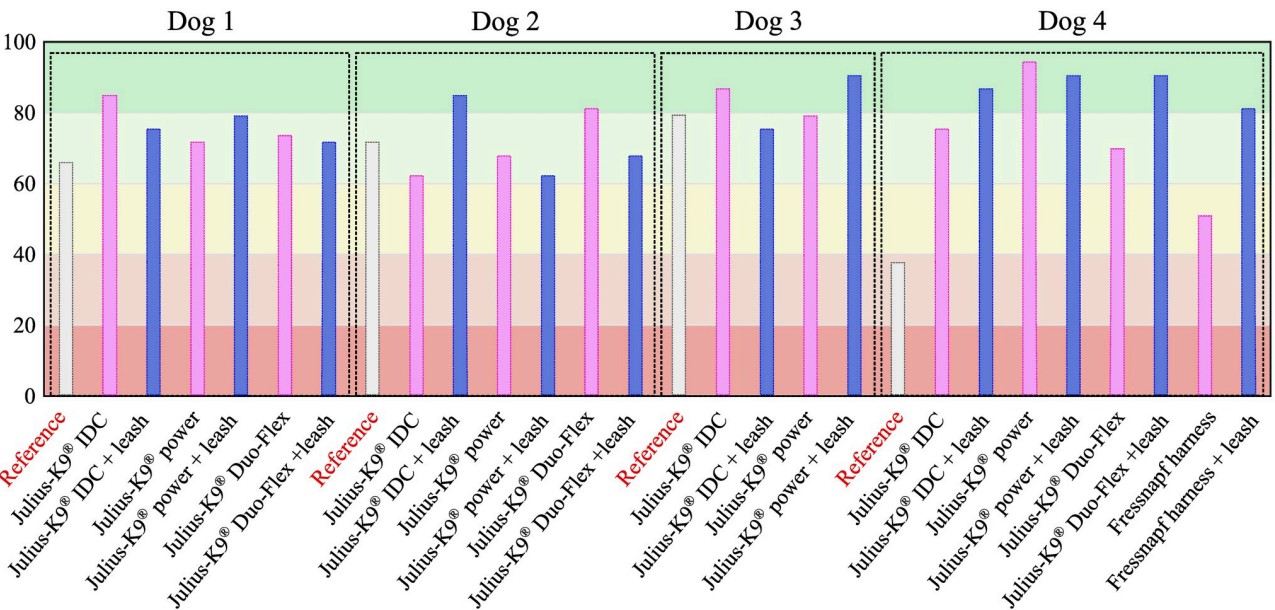

**Fig 10. Result of Anderson-Darling normality test for each trial.** Percentage of parameters of a given trial that passed the Anderson-Darling normality test ($p > 0.05$)(53 parameters in total).

The table of results for the two-sided Kolmogorov-Smirnov test can be found in S3 Table. Fig 11 shows in what percentage of cases each parameter passed the two-sided Kolmogorov-Smirnov test for identical distributions ($p > 0.05$). Fig 12 shows the inverse: how many parameters from a given configuration passed the Kolmogorov-Smirnov test.

Finally, the RMS errors between the average joint angles can be seen in Table 4. In two trials, a spine marker was not visible on the recordings, so the corresponding cells are left empty.

## Discussion

This pilot study's goal was to establish a measurement method capable of quantifying canine gait in detail, which can determine the spatio-temporal parameters of all four limbs, the joint angles of the major joints and spinal angles from the spatial coordinate of selected anatomical landmarks, captured throughout multiple gait cycles. Based on the markerset adapted from Hogy et al. (2013) (Fig 3), the 3D motion of 25 anatomical landmarks can be recorded with a motion capture system [22]. After determining an appropriate filtering method (a zero-lag 6th order Butterworth filter achieved with MatLab's *filtfilt* method [23], $f_c = 20$ Hz, with the non-harmonic components removed before and re-added after by fitting a trend line to the marker position components), the recording is segmented according to gait cycles by determining the heel strike and toe-off events based on the distance of the feet and the shoulder/hip joint in the direction of movement. In total, 18 joint angle curves (detailed in Fig 6 and Table 2), four plots of the paw movement in the sagittal plane and 53 scalar parameters (35 spatio-temporal parameters detailed in Fig 7 and Table 3 and 18 joint angle ROM parameters detailed in Fig 6 and Table 2) can be given for each gait cycle based on the recorded marker positions. An example report detailing these parameters can be seen in Fig 8. Based on Fig 9, most parameters pass the test for normality in 60+% of the cases (there were 28 trials in total). The parameters failing the test more regularly are the Swing time parameters. This is likely due to the fact that swing times have minimal values (usually between 0.2 and 0.3 seconds), where the

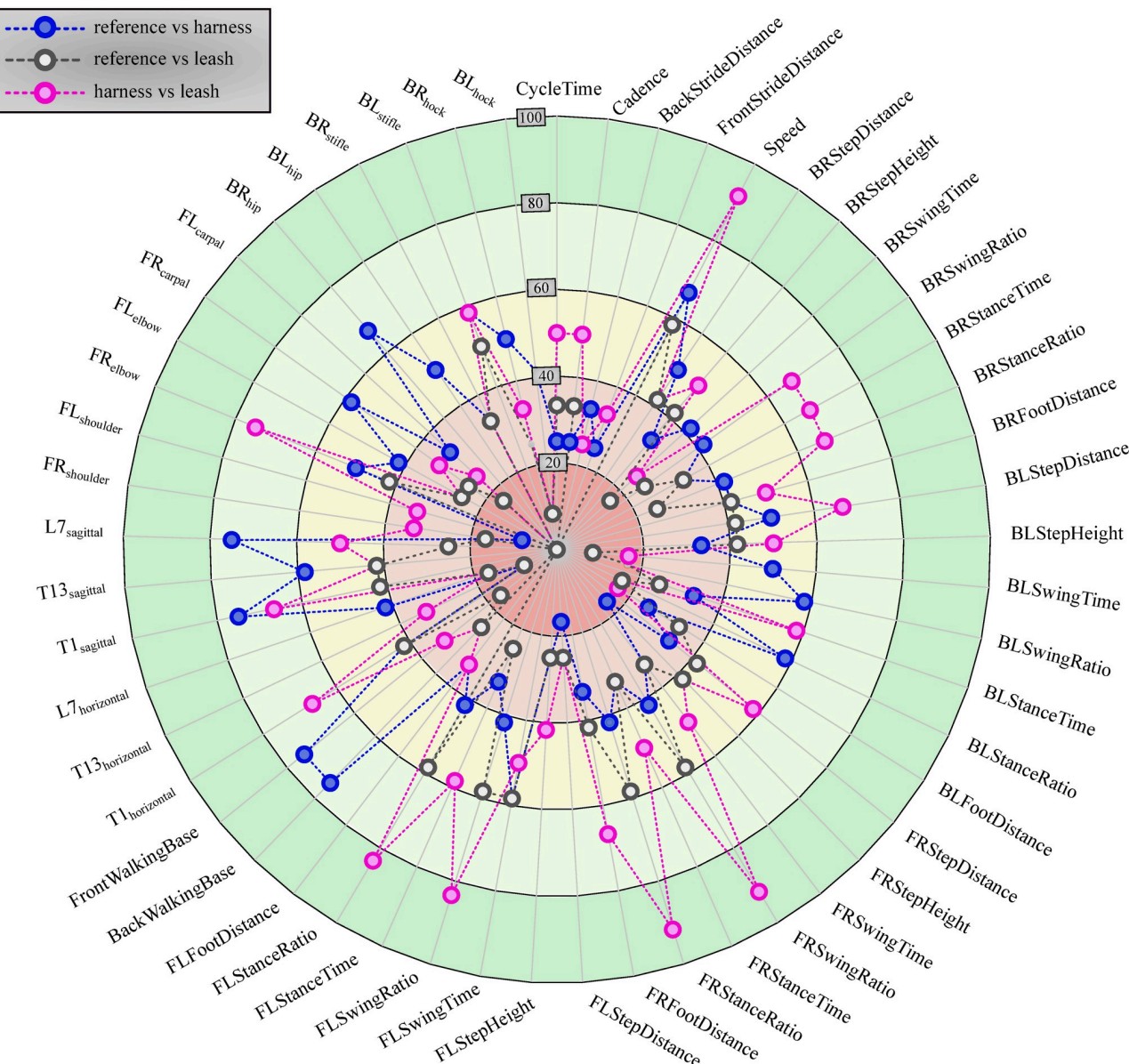

**Fig 11. Result of Kolmogorov-Smirnov test for each parameter grouped according to the different configurations.** Percentage of cases (indicated by the distance from the centre) a given calculated parameter (spatio-temporal or joint range of motion, as named on the edge of the circle) passed the two-sided Kolmogorov-Smirnov test for identical distributions ($p > 0.05$) (12 cases for each configuration in total).

variation appears to have discreet values determined by the camera's frame rate (the Anderson-Darling test is more geared towards practically continuous distributions) (Fig 13a).

Front Right Swing Time, Swing Ratio, Stance Time and Stance Ratio all failed to pass the test in case of dog 2. Examining these parameters show a few cases where swing and stance time became negative values generating outliers in the data. The outliers in stride and swing times also cause outliers in the same ratio parameters (Fig 13b). This indicates irregularities on the dog's gait where the front left heel-strike event of the amble (typically very shortly after the back right heel strike) happened too early in some cycles, before the heel strike of the back

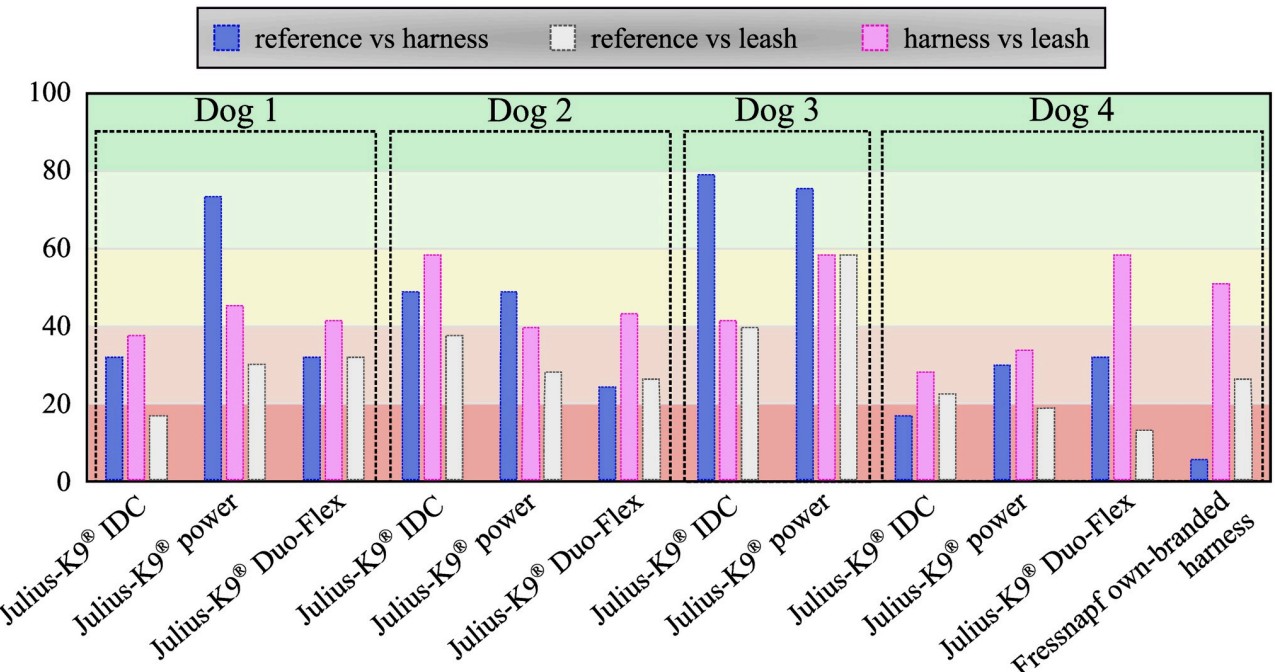

**Fig 12. Result of Kolmogorov-Smirnov test for each trial grouped according to the different configurations.** Percentage of parameters of a given case passed the two-sided Kolmogorov Smirnov test for identical distribution ($p > 0.05$)(53 parameters in total).

right limb. This indicates possible injury or other type of irregularity in the front right limb of Dog 2.

Another parameter that often does not follow normal distribution is the ROM of the sagittal aspect of the spinal angle at the T1 vertebrae. This is due to the dogs raising or lowering their heads for several gait cycles at a time as they are looking at their owner with the treats in front of them, generating larger ROM values than usual (Fig 13c).

As for the per case comparison, 60–80% of the parameters passed the normality test in most cases. The most notable case is the reference trial for dog 4, where less than 40% of the parameters passed, like the Cycle time, stride length, speed parameters. In this case, several gait cycles appeared to be much shorter in time and step lengths than the rest (Fig 13d), affecting several other parameters too, and creating long tails on one end of the distributions. No such behaviour was displayed by the same dog when in any harness so the reason for this is unclear.

The developed canine gait analysis methodology was tested in comparing the effects of different designs of harnesses on gait (which can be characterised by the 53 scalar and 18 joint angle curve and 4 paw movement plot parameters described previously) and to study whether certain designs are better than others. Based on Fig 11, apart from speed, all parameters were affected in most cases (speed being constrained by the treadmill in this case). As expected, when comparing the reference to the leashed cases, the parameters pass the two sample Kolmogorov-Smirnov test in even fewer cases. Fig 12 is a good demonstration that all of the tested harnesses alter a large portion of the calculated gait parameters for each dog tested, and as such, none of them proved to be a 'perfect harness'. Despite this, a conclusion regarding each dog can be drawn as to which harness could be preferred. For the smallest dog (Dog 3), the Julius-K9®power harness seems to be the better fit. For Dog 4, the fastest walking dog, Duo-flex seems to be the better option in leash-less cases, and the Fressnapf own-branded harness

for walking with a leash. In the case of Dog 2, IDC is the better choice, while for Dog 1, the Julius-K9®power harness. The results do not indicate any connection between the preferable harness and the breed, size or walking speed of the dogs, nor would it be realistic to draw these conclusions based on the sample size of this study.

Table 4 shows two things. Firstly, as expected the highest deviation from the natural (reference motion) is in the thoracic limbs' angles. In the most extreme cases of Dog 1 and Dog 4 (Duo-Flex cases), the large RMS errors result from the whole joint angle curve getting offset from the original (Fig 14). This is due to some markers having to be placed on top of the harness instead of directly on the dog. The positions of these markers compared to the actual location of the anatomical landmark is highly uncertain. Leash force could result in additional displacement. It has been previously shown for human gait analysis that deviations in marker placement results in similar offset errors in joint angles, but the general characteristic of the curve is unchanged [28].

A second observation based on Table 4 is that RMS errors of harness+leash cases are larger than just the harness cases. This applies for all joint angles, not just the ones potentially effected by the offset error from putting the marker on the harness. Based on the observations, the RMS error of joint angles numerically support the two intuitively known facts: harnesses

**Table 4. RMS error values for average joint angle differences.**

| Case | T1 hor. | T13 hor. | L7 hor. | T1 sag. | T13 sag. | L7 sag. | FR sho. | FL sho. | FR elb. | FL elb. | FR car. | FL car. | BR hip | BL hip | BR sti. | BL sti. | BR hock | BL hock |
|------|------|------|------|------|------|------|------|------|------|------|------|------|------|------|------|------|------|------|
| Dog 1 K9 power | 1.08 | 1.88 | 0.83 | 5.94 | 0.68 | 0.90 | 5.38 | 6.47 | 3.13 | 5.62 | 1.95 | 1.72 | 0.50 | 0.70 | 0.99 | 1.31 | 0.56 | 0.96 |
| Dog 1 K9 power (leash) | 3.41 | 4.20 | 4.04 | 16.48 | 6.42 | 3.25 | 7.09 | 10.22 | 8.23 | 9.95 | 15.73 | 12.13 | 2.59 | 2.94 | 4.14 | 3.86 | 7.35 | 6.16 |
| Dog 1 K9 IDC | 3.68 | 2.72 | 1.34 | 8.20 | 1.58 | 0.55 | 8.16 | 21.85 | 2.65 | 21.96 | 3.17 | 4.57 | 1.30 | 1.78 | 2.24 | 2.88 | 2.09 | 2.59 |
| Dog 1 K9 IDC (leash) | 1.79 | 3.61 | 4.13 | 13.23 | 5.91 | 3.13 | 17.90 | 13.87 | 16.79 | 14.06 | 22.60 | 22.73 | 2.74 | 2.32 | 7.22 | 5.77 | 11.66 | 8.18 |
| Dog 1 K9 Duo-Flex | 1.26 | 1.44 | 1.14 | 3.19 | 2.60 | 0.39 | 37.22 | 38.37 | 21.70 | 22.86 | 6.76 | 5.19 | 1.43 | 1.24 | 1.91 | 1.87 | 2.78 | 2.33 |
| Dog 1 K9 Duo-Flex (leash) | 1.95 | 2.23 | 4.42 | 10.00 | 6.65 | 6.74 | 63.26 | 63.75 | 35.63 | 33.16 | 15.50 | 14.04 | 3.11 | 3.22 | 9.54 | 6.98 | 10.89 | 7.96 |
| Dog 2 K9 power | 2.00 | 7.63 | 5.56 | 23.56 | 6.83 | 1.07 | 10.09 | 3.73 | 9.63 | 13.52 | 7.04 | 4.38 | 3.35 | 3.11 | 1.29 | 1.60 | 1.31 | 1.13 |
| Dog 2 K9 power (leash) | 5.53 | 9.42 | | 17.04 | 10.96 | | 7.54 | 4.62 | 5.39 | 4.68 | 7.25 | 6.40 | 5.41 | 11.10 | 6.43 | 9.26 | 6.01 | 14.32 |
| Dog 2 K9 IDC | 2.41 | 3.64 | 2.97 | 9.31 | 0.72 | 0.55 | 15.57 | 5.08 | 17.18 | 8.94 | 5.67 | 6.02 | 1.14 | 1.57 | 0.97 | 1.26 | 1.81 | 2.34 |
| Dog 2 K9 IDC (leash) | 3.23 | 5.55 | 5.04 | 20.54 | 5.88 | 3.21 | 12.11 | 6.89 | 16.70 | 11.87 | 5.97 | 6.76 | 4.00 | 6.82 | 3.82 | 4.14 | 4.92 | 7.42 |
| Dog 2 K9 Duo-Flex | 2.12 | 2.80 | 1.74 | 11.86 | 1.90 | 0.58 | 15.47 | 8.84 | 15.60 | 13.24 | 8.97 | 10.90 | 3.12 | 3.24 | 2.28 | 2.01 | 3.88 | 3.80 |
| Dog 2 K9 Duo-Flex (leash) | 4.45 | 11.75 | 7.95 | 14.63 | 5.52 | 5.03 | 7.63 | 8.58 | 10.10 | 10.49 | 13.34 | 15.11 | 5.83 | 12.90 | 6.60 | 6.55 | 7.36 | 8.75 |
| Dog 3 K9 power | 1.76 | 5.64 | 2.26 | 1.67 | 4.86 | 0.99 | 5.80 | 11.58 | 8.73 | 7.63 | 10.04 | 6.70 | 0.88 | 0.78 | 1.68 | 2.51 | 1.99 | 2.67 |
| Dog 3 K9 power (leash) | 5.24 | 10.52 | | 3.97 | 22.15 | | 11.98 | 14.01 | 18.30 | 14.00 | 36.71 | 31.34 | 3.01 | 2.69 | 3.28 | 3.91 | 4.09 | 4.50 |
| Dog 3 K9 IDC | 2.59 | 6.71 | 5.54 | 3.32 | 0.91 | 0.89 | 6.41 | 6.10 | 6.82 | 6.68 | 9.43 | 7.99 | 1.00 | 0.72 | 1.15 | 1.45 | 1.04 | 1.47 |
| Dog 3 K9 IDC (leash) | 4.46 | 9.02 | 3.79 | 6.64 | 4.03 | 2.90 | 10.76 | 6.69 | 9.09 | 5.93 | 16.47 | 13.00 | 3.98 | 4.37 | 8.18 | 10.10 | 8.61 | 9.68 |
| Dog 4 K9 power | 3.72 | 1.96 | 1.55 | 7.19 | 1.50 | 0.89 | 10.11 | 17.18 | 18.66 | 24.19 | 15.26 | 21.46 | 2.12 | 1.47 | 3.05 | 3.30 | 4.42 | 3.64 |
| Dog 4 K9 power (leash) | 2.69 | 2.26 | 2.99 | 11.80 | 1.54 | 2.67 | 6.38 | 21.34 | 24.51 | 33.84 | 13.31 | 14.89 | 1.71 | 1.79 | 6.47 | 6.40 | 7.72 | 6.83 |
| Dog 4 K9 IDC | 2.85 | 2.32 | 0.76 | 3.83 | 1.66 | 1.00 | 9.77 | 7.85 | 6.84 | 5.07 | 21.23 | 12.31 | 2.98 | 2.27 | 3.85 | 3.72 | 5.04 | 4.21 |
| Dog 4 K9 IDC (leash) | 2.68 | 4.84 | 3.68 | 5.11 | 1.58 | 1.57 | 12.81 | 15.41 | 16.81 | 26.18 | 19.48 | 11.78 | 2.35 | 1.67 | 5.12 | 5.76 | 6.19 | 6.06 |
| Dog 4 K9 Duo-Flex | 2.33 | 1.21 | 1.02 | 3.96 | 1.63 | 1.09 | 46.08 | 57.44 | 23.27 | 34.83 | 12.80 | 13.26 | 1.09 | 0.87 | 1.64 | 1.82 | 1.60 | 1.74 |
| Dog 4 K9 Duo-Flex (leash) | 2.21 | 1.63 | 2.93 | 9.32 | 3.02 | 4.21 | 68.75 | 79.10 | 37.14 | 48.67 | 10.83 | 10.85 | 1.77 | 1.55 | 4.50 | 6.14 | 5.41 | 6.25 |
| Dog 4 Fressnapf | 11.32 | 2.16 | 3.71 | 11.88 | 1.44 | 5.79 | 15.65 | 4.63 | 6.83 | 3.42 | 5.85 | 14.36 | 5.69 | 11.31 | 7.40 | 6.21 | 11.00 | 16.15 |
| Dog 4 Fressnapf (leash) | 8.16 | 4.62 | 1.61 | 9.63 | 2.24 | 9.45 | 27.26 | 11.87 | 6.37 | 7.15 | 7.53 | 14.39 | 6.09 | 12.43 | 3.97 | 3.83 | 15.83 | 21.98 |

Values are in degrees. Colours from green through yellow to red indicate lowest to highest RMS. Missing values indicate joint angles that could not be calculated for that case.

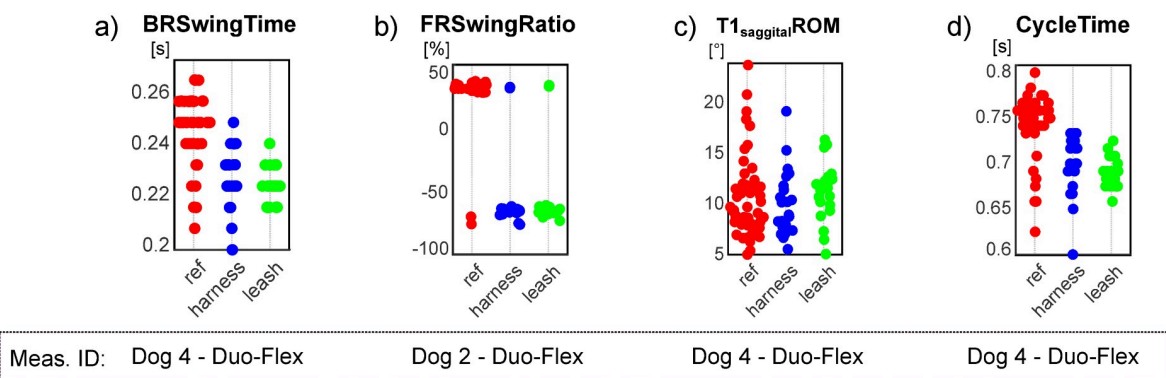

**Fig 13. Anomalistic distributions.** a) discreet values for swing time; b) negative swing ratio in case of Dog 2; c) distribution of T1 sagittal spine angle range of motion with a notable tails towards larger values; d) large tail towards lower values on the cycle time parameter of Dog 4 on the reference trial (red points).

mostly affect thoraic limbs more, and the addition of a leash further deviates the motion from the natural one.

Comparing the results with the study of Lafuente et al., the results can be said to agree [13]. Fig 11 shows that ROM of the shoulder joint was significantly affected by the inclusion of a harness compared to the reference case in virtually all trials, with the harness and leash cases showing smaller number of significantly different cases compared to harness only trials. Table 4 shows that the non-restrictive style harness (Duo-flex) shows significantly larger change on the shoulder joint angles compared to the other harnesses for dogs 1 and 4, while showing about the same for dog 2 (dog 3 was not tested in the duo-flex harness). This also agrees with the findings of Lafuente et al. stating that the non-restrictive harnesses are actually more restrictive in respect to the shoulder angles. However, as stated before these cases are the ones markers might have had to been placed on top of the marker, causing high uncertainty. Comparing the results with the study of Peham et al. [16], Fig 11 also confirms that harnesses significantly affect the movement of the spine, although the effect seems to be more pronounced in the horizontal plane compared to the sagittal motions.

To test the possible limitations of this method, four dogs with varying body shapes and sizes were measured comparing a number of different harnesses. The studied dogs did not provide enough data for any meaningful conclusions on evaluating different harnesses. Combined

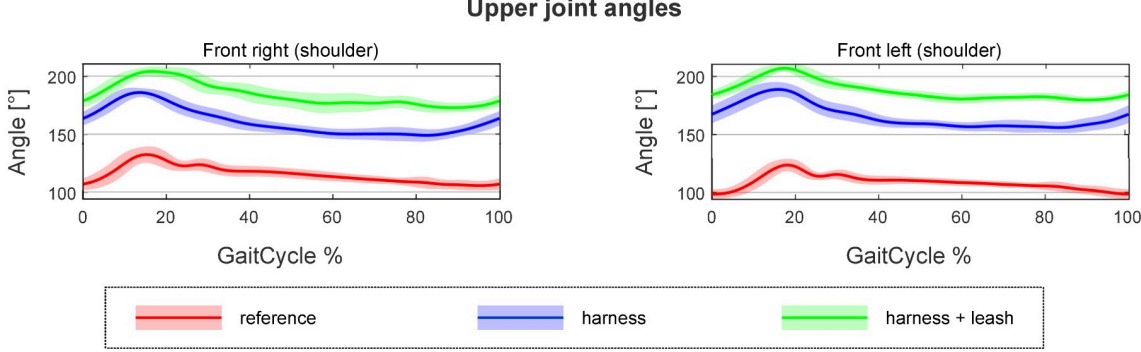

**Fig 14. Offset in the shoulder angles in the case of dog 4 Duo-Flex.**

with the large number of parameters calculated, the amount of data is far less than necessary to draw clear conclusions about how individual parameters are affected. The difference in breeds and body sizes of the dogs further hinder drawing these conclusions. However, it was shown that the method itself is usable regardless of the body size of the dog, or the type of the harness.

A limitation of the method is using projected 2D joint angles instead of anatomically defined 3D joint angles (flexion-extension, abduction-adduction, internal-external rotation) for the limbs. Proper calculation of these angles require information about the 3D rigid-body motion of each segment of the limbs, which in turn require at least 3 markers on each body segment. While this would be plausible for larger dogs, in case of small dogs the markers would have to be too close for each other for our current setup. The possibility of anatomically defined joint angles on all sizes of dogs could be further explored with a more specialised setup, where the cameras are placed in the immediate surroundings of the treadmill. 2D joint angles are also more comparable with single camera measurements, where joint angles are determined on a single video recording of the dog from the side.

An other issue with examining harnesses specifically, is that some harnesses might end up covering an anatomical landmark of the dog (the FR2 anatomical landmark in most cases). In this study, we chose to place the marker on top of the harness in these cases, as close to the the landmark below as possible. However, this adds an extra layer on top of the skin that can shift compared to the landmark, causing the results to be inaccurate, and affecting the calculation of the front shoulder and elbow joint angles.

We aim to continue this research in a larger MoCap studio, where we can test more dogs within several groups of body sizes performing walk and trot on the ground instead of a treadmill to observe more natural movements. The anomalistic parameter distribution presented on Fig 13 should also be further studied, particularly in the case of getting negative values for the swing time and ratio parameters.

## Conclusion

The novelty of this study is the large number of computable gait parameters established for canine gait analysis: spatio-temporal parameters for all four limbs, the sagittal aspect of the major limb joint angles, and the sagittal and horizontal aspects of spinal angles, as well as the path of any measured anatomical landmark (paw movement in the sagittal plane was highlighted here). An appropriate filtering process for the marker coordinates in case of dogs were also established. These calculated gait parameters were examined for four different dogs, comparing cases of movement on a treadmill in a natural and harness-wearing (with and without leash) state. This pilot study demonstrates that 3D gait analysis presents an opportunity for examining an extensive array of parameters and providing in-depth analysis for multiple purposes, whether it would be for examining the movements of a single dog for veterinary treatment or training, or studying the effects of different harness designs, with a more extensive parameter set than before. The results demonstrate that the method is applicable to various dog sizes and harnesses.

The results of the Kolmogorov-Smirnov test indicate that the least similar gait pattern is obtained when comparing the reference to the leashed cases. However, the tendencies based on the results show that all the tested harness altered the gait pattern for each dog and such there would be hard to find a one-size-fits-all option from them. For example some harnesses might perform better on smaller size dogs, while a different harness might be better suited for larger ones, and that is not even considering the different use cases, i.e., a walk in the city compared with a hike in the woods, or work for a service dog. This is an empirically accepted standpoint, but there is no research supporting it as of yet to the authors' knowledge. More

specific findings regarding the "goodness" of the harnesses would requires measurements with more participating dogs, which would be one of the possible advances in this present research. The method presented here can be utilised to evaluate harnesses on a dog-by-dog cases, selecting the most suitable for the given activities. The method can also be re-purposed to any case, where multiple sets of canine gaits need to be compared.

## Supporting information

**S1 Table. Table of calculated scalar parameters.** Tabular results of the 53 calculated scalar parameters for all measurement scenarios.
(PDF)

**S2 Table. Tabular results of the Anderson-Darling normality test.** Tabular results of the p-values of the Anderson-Darling normality test for the 53 calculated scalar parameters for all measurement scenarios.
(PDF)

**S3 Table. Tabular results of the two-sided Kolmogorov-Smirnov test.** Tabular results of the p-values of the two-sided Kolmogorov-Smirnov test for the 53 calculated scalar parameters comparing reference and harness trials, harness and harness+leash trials and reference and harness+leash trials.
(PDF)

## Acknowledgments

The authors would like to thank Dr. Otília Biksi for her contribution in overseeing the experiments.

## Author Contributions

**Conceptualization:** Zsófia Pálya, Kristóf Rácz, Gergely Nagymáté, Rita M. Kiss.

**Data curation:** Gergely Nagymáté.

**Investigation:** Zsófia Pálya, Kristóf Rácz, Gergely Nagymáté, Rita M. Kiss.

**Methodology:** Zsófia Pálya, Kristóf Rácz, Gergely Nagymáté, Rita M. Kiss.

**Software:** Kristóf Rácz.

**Supervision:** Rita M. Kiss.

**Visualization:** Zsófia Pálya, Rita M. Kiss.

**Writing – original draft:** Zsófia Pálya, Kristóf Rácz, Gergely Nagymáté, Rita M. Kiss.

**Writing – review & editing:** Zsófia Pálya, Kristóf Rácz, Gergely Nagymáté, Rita M. Kiss.

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
