## [Decision Letter · Decision Letter 0]

9 Sep 2021

PONE-D-21-02564Developing a detailed canine gait analysis method for evaluating harnesses: a pilot studyPLOS ONE

Dear Dr. Rita M. Kiss,

Thank you for submitting your manuscript to PLOS ONE. After careful consideration, we feel that it has merit but does not fully meet PLOS ONE’s publication criteria as it currently stands. Therefore, we invite you to submit a revised version of the manuscript that addresses the points raised during the review process.

We look forward to receiving your revised manuscript.

Kind regards,

Ewa Tomaszewska, DVM Ph.D

Academic Editor

PLOS ONE

Additional Editor Comments (if provided):

Dear Authors,

In Reviewer`s opinion, the global quality of this study is ABOVE AVERAGE, but should be corrected in accordance to comments sent by Reviewer.

Journal Requirements:

2. In your Methods section, please provide additional details regarding participant consent from the owners of the animals. In the ethics statement in the Methods and online submission information, please ensure that you have specified (1) whether consent was informed and (2) what type you obtained (for instance, written or verbal). If the need for consent was waived by the ethics committee, please include this information.

3. We note that Figures 1,2,3 and 4 in your submission contain copyrighted images. All PLOS content is published under the Creative Commons Attribution License (CC BY 4.0), which means that the manuscript, images, and Supporting Information files will be freely available online, and any third party is permitted to access, download, copy, distribute, and use these materials in any way, even commercially, with proper attribution. For more information, see our copyright guidelines: http://journals.plos.org/plosone/s/licenses-and-copyright.

a. You may seek permission from the original copyright holder of Figures 1,2,3 and 4 to publish the content specifically under the CC BY 4.0 license. 

Reviewers' comments:

Reviewer's Responses to Questions

**Comments to the Author**

1. Is the manuscript technically sound, and do the data support the conclusions?

Reviewer #1: Partly

2. Has the statistical analysis been performed appropriately and rigorously? 

Reviewer #1: Yes

3. Have the authors made all data underlying the findings in their manuscript fully available?

Reviewer #1: Yes

4. Is the manuscript presented in an intelligible fashion and written in standard English?

Reviewer #1: Yes

5. Review Comments to the Author

Reviewer #1: This study sought to present a detailed method for canine gait acquisition and analysis addressing the need to correctly assess dog harnesses, which should adapt to the breed, size, and use scenarios. In particular, the authors hypothesized that the presence of the harnesses can alter the canine gait. For this reason, the author implemented this experimental pilot study to quantify how different harnesses affected the dogs’ kinematics during walking with respect to the unleashed condition. Four trained dogs and different harnesses (three provided by Juliuus-K9 and a one custom-made) with leash and without leash were included in the analysis. The study was approved by an Ethical Committee in 2015. The acquisition protocol was based on the use of 25 markers and allowed to estimate 18 joint angles, 4 paw movement paths in the sagittal plane and a wide set of spatio-temporal parameters, which can be normalized on multiple gait cycles. The authors reported information about marker trajectories processing and results are then reported with respect to the identified breeds and harnesses. The authors claimed that the methods they proposed can be used in comparative assessments; furthermore, they reported that all the analysed harnesses altered dogs kinematics during plain walking on treadmill, but the system was able to provide useful information about a possible optimal choice. All the data are available without restriction.

General Comment

The hypothesis at the basis of this paper is clearly reported as far as the main objective. Both the experimental phase and data analysis are written with a good level of details; the analysis in particular was performed very well, and the synthesis of the obtained results is extremely appreciable. Although the methodology in itself is not that innovative, the application to canine movements provides useful hints and novel perspective.

The structure of the article seems to be precise (Abstract, Introduction, Methodology [with subheadings], Results, Discussion).

Experimental phase and data analysis seem to be clearly reported and are coherent with the work objectives. Several minor concerns are hereinafter reported.

The use of the English language seems to be correct.

Specific Comments

Title

Ok. I would only consider to change “developing” with “development of”.

Abstract

In general, this section is quite ok. Please, could you report some quantitative information about the most interesting findings you obtained. It is important to get – even from the abstract – why you thought that your method is reliable and the differences you were able to highlight.

Introduction

In general also this section is ok; I would only state better the main limitations of the actual studies and the main novelty and innovation of the methodology you proposed, not only in light of the specific application you dealt with (i.e., harnesses).

Methods

• Page 3/15 Line 79-80: Please justify the number of dogs you involved in this study; although this is a pilot study, the reader needs further information about the choice of the canine subjects and how this can affect the possible generalization of the main findings (to be discuss in the Discussion section).

• Page 3 Line 78. A period “.” seems to miss at the end of the sentence.

• Page 3 Table 1. Please provide further information about the gait patterns.

• Page 4/15 Line 91: Please give further information about the third-party harness. Why did you call it “third-party”? Were Julius-K9s’ ones not third-party harnesses? In figure 1 there is no representation of this forth harness.

• Page 4/15 Line 101: I think that there is a typo here “[refence: Motive]”, and a correctly reported reference.

• Page 4/15 Line 103: Please provide numerical information about the accuracy you obtained in your specific setup after volume calibration.

• Page 4/15 Line 122-124: Please justify the choice of placing the marker on the harness here or discuss any possible issue in the Discussion section. Harness could significantly move with respect to the dog’s body, couldn’t it?

• Page 4/15 Line 127: Please provide more detail about the “clean up” phase.

• Page 5/15 Line 183: Please justify the use of 2D projections, or otherwise, discuss this as a possibile actual limitation of this study.

Results

Very well reported both graphically and in the main text.

Discussion

In general, this section is ok since it discuss your main findings. However is quite missing a comparison with the current literature, at least, where information are available. Furthermore, you tried to underline possible limitation at page 12/15 line 331-338, but you have to underline better any issue that can limit the possible generalization of your results (including the methodological ones, as – for instance – the choice of estimating 2D joint angles).

Conclusions

• Page 13/15 Line 359-362: This sentence can be hardly supported by the results of this only study. If there are differences, with such a small sample size it is not quite possible to reliably ascribe them to the only type of harnesses.

References

The references to previous works seem to be precise, wide and up-to-date.

Figures

Very good.

Tables

Good.

6. PLOS authors have the option to publish the peer review history of their article (what does this mean?). If published, this will include your full peer review and any attached files.

Reviewer #1: **Yes: **Nicola Francesco Lopomo

---

## [Author Response · Author response to Decision Letter 0]

5 Oct 2021

Respose to Reviewer

Dear Dr. Lopomo,

We have received your review of your article, and we would like to thank you for your extraordinarily thorough and relevant questions and suggestions! In the following section we will address each point you raised to the best of our abilities. Please find the full response with all the listed modifications in the attached PDF!

Title:

I would only consider to change “developing” with “development of”.

We do agree with this suggestion, and modified the title accordingly!

Abstract:

In general, this section is quite ok. Please, could you report some quantitative information about the most interesting findings you obtained. It is important to get – even from the abstract – why you thought that your method is reliable and the differences you were able to highlight.

Thank you for the insightful comment. We have extended the abstract with some more detail and qualitative results of the statistical analysis. Also, we‘ve reworded the beginning not to exceed 300 words.

Introduction:

In general also this section is ok; I would only state better the main limitations of the actual studies and the main novelty and innovation of the methodology you proposed, not only in light of the specific application you dealt with (i.e., harnesses).

Thank you for briging our attention tot his point! We have changed the wording of the last paragraph of the introduction, to better reflect the novelty of our study, which is the level of complexity in the analysed parameters, and to point out that this method can also be used in research of canine motion that does not deal with harnesses!

Methods:

1. Page 3/15 Line 79-80: Please justify the number of dogs you involved in this study; although this is a pilot study, the reader needs further information about the choice of the canine subjects and how this can affect the possible generalization of the main findings (to be discuss in the Discussion section).

We have no other justification for it other than that’s how many we could find whose owners were willing to participate, and train their dogs beforehand for walking on a treadmill. We changed the wording for this to be more clearer in the paper.

2. Page 3 Line 78. A period “.” seems to miss at the end of the sentence.

Thank you, we added the missing period!

3. Page 3 Table 1. Please provide further information about the gait patterns.

We have added a reference with detailed descriptions of every canine gait pattern. 

(Table 1. Participating dogs and used harnesses. Gait patterns were identified by eye by an expert. Detailed description of each gait pattern can be found in [18]. 

[...]

18. Zink C, Carr BJ. Locomotion and Athletic Performance. In: Zink C, Van DykeJB, editors. Canine Sports Medicine and Rehabilitation, Second Edition. NewYork, USA: John Wiley & Sons, Inc; 2018. pp. 23–42.)

4.Page 4/15 Line 91: Please give further information about the third-party harness. Why did you call it “third-party”? Were Julius-K9s’ ones not third-party harnesses? In figure 1 there is no representation of this forth harness.

The owner of dog 4 brought this harness to the measurement, and it was a generic non-restrictive type. Since the owner had no information about the manufacturer, we were not able to name any. This also means that we do not have a right to publish any picture of it. One for sure, it was not a Julis-K9 manufactured harness. Thank you for bringing this shortcoming to our attention; we have extended the description in the Methods section.

5. Page 4/15 Line 101: I think that there is a typo here “[refence: Motive]”, and a correctly reported reference.

Thank you for this remark. Further information about the calibration accuracy has been added to this section.

6. Page 4/15 Line 122-124: Please justify the choice of placing the marker on the harness here or discuss any possible issue in the Discussion section. Harness could significantly move with respect to the dog’s body, couldn’t it?

We strongly agree that this point should have been brought up in the discussion as a limitation of the study from the beginning. The harness can indeed move compared to the body considerably, but unfortunately in cases where the harness covered an anatomical landmark, we had no other way to get approximate position data of the given anatomical landmark (FR2) with the given measurement setup.

7. Page 4/15 Line 127: Please provide more detail about the “clean up” phase.

Unfortunately our previous wording made it look like there was some additional clean up that took place before labelling and exporting relevant sections of data, when in reality the labelling and exporting process was the “clean up”. Thank you for pointing out the ambiguous wording of this subsection! We have rewritten it to more clearly describe the process.

(A technician first processed the recorded marker data in Motive as follows: markers were labelled according to the used marker-set (Fig. 3) for each recording. Next, a section of homogeneous gait between receiving treats was selected for each trial, and exported into a text file containing metadata of the measurement in a header – like frame rate and130total number of frames – and the marker position data for each frame. For all further calculations, MATLAB (R2020b) was used [16].)

8. Page 5/15 Line 183: Please justify the use of 2D projections, or otherwise, discuss this as a possibile actual limitation of this study.

While we would have liked to used proper anatomical joint angles, the marker configuration required for it would have been much more complex, and very impractical on the smaller dogs with the camera system of our laboratory. We have added a paragraph to the discussion about this limitation explaining the requirements for calculating 3D join angles!

(A limitation of the method is using projected 2D joint angles instead of anatomically defined 3D joint angles (flexion-extension, abduction-adduction, internal-external rotation) for the limbs. Proper calculation of these angles require information about the 3D rigid-body motion of each segment of the limbs, which in turn require at least 3 markers on each body segment. While this would be plausible for larger dogs, in case of small dogs the markers would have to be too close for each other for our current setup. The possibility of anatomically defined joint angles on all sizes of dogs could be further explored with a more specialised setup, where the cameras are placed in the immediate surroundings of the treadmill. 2D joint angles are also more comparable with single camera measurements, where joint angles are determined on a single video recording of the dog from the side.)

Discussion:

In general, this section is ok since it discuss your main findings. However is quite missing a comparison with the current literature, at least, where information are available. Furthermore, you tried to underline possible limitation at page 12/15 line 331-338, but you have to underline better any issue that can limit the possible generalization of your results (including the methodological ones, as – for instance – the choice of estimating 2D joint angles).

Thank you for pointing out the missing comparison with the literature! Although not a lot of data can be found on the effects of harnesses, those we found do not contradict our results. A section has been added to the discussion comparing the results.

The limitations noted in previous suggestions (2D projections of joint angles and marker placement on harness) has also been addressed in the discussion.

Conclusion:

Page 13/15 Line 359-362: This sentence can be hardly supported by the results of this only study. If there are differences, with such a small sample size it is not quite possible to reliably ascribe them to the only type of harnesses.

Thank you for this observation. We have changed the wording of the last paragraph of the Conclusion, and the statement improved, as can be supported by our results accordingly. Some possible advances of the study have also been added to the paragraph.

Based on your comments and suggestions we have revised our manuscript, and we hope that our revisions prove to be satisfactory! Thank you again for your insightful critique. It helped us greatly in bringing our research up to par with the standards expected by the scientific community!

Best regards:

Zsófia Pálya, Kristóf Rácz, Gergely Nagymáté & Rita M. Kiss

---

## [Decision Letter · Decision Letter 1]

27 Oct 2021

PONE-D-21-02564R1Development of a detailed canine gait analysis method for evaluating harnesses: a pilot studyPLOS ONE

Dear Dr. Rita M. Kiss,

Thank you for submitting your manuscript to PLOS ONE. After careful consideration, we feel that it has merit but does not fully meet PLOS ONE’s publication criteria as it currently stands. Therefore, we invite you to submit a revised version of the manuscript that addresses the points raised during the review process.

We look forward to receiving your revised manuscript.

Kind regards,

Ewa Tomaszewska, DVM Ph.D

Academic Editor

PLOS ONE

Journal Requirements:

Reviewers' comments:

Reviewer's Responses to Questions

**Comments to the Author**

1. If the authors have adequately addressed your comments raised in a previous round of review and you feel that this manuscript is now acceptable for publication, you may indicate that here to bypass the “Comments to the Author” section, enter your conflict of interest statement in the “Confidential to Editor” section, and submit your "Accept" recommendation.

Reviewer #1: (No Response)

2. Is the manuscript technically sound, and do the data support the conclusions?

Reviewer #1: Partly

3. Has the statistical analysis been performed appropriately and rigorously? 

Reviewer #1: Yes

4. Have the authors made all data underlying the findings in their manuscript fully available?

Reviewer #1: Yes

5. Is the manuscript presented in an intelligible fashion and written in standard English?

Reviewer #1: Yes

6. Review Comments to the Author

Reviewer #1: I would like to thank the authors for the great effort they realized to answer all the concerns arisen during the first round of review.

However, several issues are still open and need further clarification:

Question: 1. Page 3/15 Line 79-80: Please justify the number of dogs you involved in this study; although this is a pilot study, the reader needs further information about the choice of the canine subjects and how this can affect the possible generalization of the main findings (to be discuss in the Discussion section).

Answer: We have no other justification for it other than that’s how many we could find whose owners were willing to participate, and train their dogs beforehand for walking on a treadmill. We changed the wording for this to be more clearer in the paper.

Further Comment: I guess that the choice of the number and breeds of the dogs should be justified better; a scientific approach requires that the sample size and characteristics must be defined before recruitment starts. Since the number of dog is extremely reduced and this could impact the possibility to generalize your approach, please state at your best the hypotheses at the basis of your choice.

Question: 4.Page 4/15 Line 91: Please give further information about the third-party harness. Why did you call it “third-party”? Were Julius-K9s’ ones not third-party harnesses? In figure 1 there is no representation of this forth harness.

Answer: The owner of dog 4 brought this harness to the measurement, and it was a generic non-restrictive type. Since the owner had no information about the manufacturer, we were not able to name any. This also means that we do not have a right to publish any picture of it. One for sure, it was not a Julis-K9 manufactured harness. Thank you for bringing this shortcoming to our attention; we have extended the description in the Methods section.

Further Comment: Unfortunately you did not get my hint. I underlined the fact that your study, although you reported that it is not, seems to be sponsored by Julius-K9 (as you reported, however, the three harnesses from Julius-K9 was provided free of charge by the manufacturer...that is a sort of sponsorship). For sure, you can, but you have to report better the main hypotheses at the basis of your research and why you did chose only Julus-K9 harnesses and, more specifically, those models. Since this is not a sponsored study, the 3rd party harness should be treated as those ones provided by Julius-K9, maybe using it as reference. You can provide a wider description and a sketch of this harness, if useful to better understand your results.

7. PLOS authors have the option to publish the peer review history of their article (what does this mean?). If published, this will include your full peer review and any attached files.

Reviewer #1: **Yes: **Nicola Francesco Lopomo

---

## [Author Response · Author response to Decision Letter 1]

4 Jan 2022

Revision Round 2:

Dear Dr. Lopomo,

We have studied your suggestions and we would like to thank you for your constructive comments again. The responses of all the comments and recommendations are listed below.

Comment #1:

Question: Page 3/15 Line 79-80: Please justify the number of dogs you involved in this study; although this is a pilot study, the reader needs further information about the choice of the canine subjects and how this can affect the possible generalization of the main findings (to be discuss in the Discussion section).

Answer: We have no other justification for it other than that’s how many we could find whose owners were willing to participate, and train their dogs beforehand for walking on a treadmill. We changed the wording for this to be more clearer in the paper.

Further Comment: I guess that the choice of the number and breeds of the dogs should be justified better; a scientific approach requires that the sample size and characteristics must be defined before recruitment starts. Since the number of dog is extremely reduced and this could impact the possibility to generalize your approach, please state at your best the hypotheses at the basis of your choice.

We do understand the concern about the sample size of the pilot study. We did aim to have at least one small, medium and large size dog in the study to confirm the method is suitable for all sizes. Thankfully, all categories were represented with the dogs we could recruit. We added a few regards about this to the text of the manuscript.

Comment #2:

Question: 4.Page 4/15 Line 91: Please give further information about the third-party harness. Why did you call it “third-party”? Were Julius-K9s’ ones not third-party harnesses? In figure 1 there is no representation of this forth harness.

Answer: The owner of dog 4 brought this harness to the measurement, and it was a generic non-restrictive type. Since the owner had no information about the manufacturer, we were not able to name any. This also means that we do not have a right to publish any picture of it. One for sure, it was not a Julis-K9 manufactured harness. Thank you for bringing this shortcoming to our attention; we have extended the description in the Methods section.

Further Comment: Unfortunately you did not get my hint. I underlined the fact that your study, although you reported that it is not, seems to be sponsored by Julius-K9 (as you reported, however, the three harnesses from Julius-K9 was provided free of charge by the manufacturer...that is a sort of sponsorship). For sure, you can, but you have to report better the main hypotheses at the basis of your research and why you did chose only Julus-K9 harnesses and, more specifically, those models. Since this is not a sponsored study, the 3rd party harness should be treated as those ones provided by Julius-K9, maybe using it as reference. You can provide a wider description and a sketch of this harness, if useful to better understand your results.

Thank you for this insightfull remark. We added some more general regards about the selected harnesses. Moreover, we would like to point out that in the case of Julius-K9 company the production takes place in Hungary. Upon our request they offered to manufacture the examined harrnesses without the reflective elements, thus facilitating our research work. In connection with harness bring by Dog 4 owner, we were also possible to find out where it was purchased and whether it was an own-branded product. In light of this, the “3rd party” name has been changed to “Fressnapf own-branded”. Based on the measurement photos and the suggestion, a sketch of this harness was added to the Figure 1.

Based on your comments and suggestions we have revised our manuscript, and we hope that our revisions proves to be satisfactory! Thank you again for your insightful critique. It helped us greatly in bringing our research up to par with the standards expected by the scientific community!

Best regards:

Zsófia Pálya, Kristóf Rácz, Gergely Nagymáté & Rita M. Kiss

---

## [Editor Report · Decision Letter 2]

9 Feb 2022

Development of a detailed canine gait analysis method for evaluating harnesses: a pilot study

PONE-D-21-02564R2

Dear Dr. Rita M. Kiss,

We’re pleased to inform you that your manuscript has been judged scientifically suitable for publication and will be formally accepted for publication once it meets all outstanding technical requirements.

Kind regards,

Ewa Tomaszewska, DVM Ph.D

Academic Editor

PLOS ONE

---

## [Editor Report · Acceptance letter]

14 Feb 2022

PONE-D-21-02564R2 

Development of a detailed canine gait analysis method for evaluating harnesses: a pilot study 

Dear Dr. Kiss:

I'm pleased to inform you that your manuscript has been deemed suitable for publication in PLOS ONE. Congratulations! Your manuscript is now with our production department. 

Kind regards, 

on behalf of

Professor Ewa Tomaszewska 

Academic Editor

PLOS ONE